**Subject Category:**
Biology (whole organism)

health and disease and epidemiology/ microbiology/bioinformatics

coral microbes, disease, immunity, marine infrastructure, bacteria, white syndrome

**Author for correspondence:**
David G. Bourne
e-mail: david.bourne@jcu.edu.au

[†]F.J.P. and J.B.L. should be considered joint first authors.

# Reduced diversity and stability of coral-associated bacterial communities and suppressed immune function precedes disease onset in corals

F. Joseph Pollock[1,2,3,4,5,†], Joleah B. Lamb[6,†],
Jeroen A. J. M. van de Water[1,2,3,4,7], Hillary A. Smith[1],
Britta Schaffelke[3], Bette L. Willis[1,2,4]
and David G. Bourne[1,3,4]

[1]College of Science and Engineering, and [2]ARC Centre of Excellence for Coral Reef Studies, James Cook University, Townsville, Queensland, Australia
[3]Australian Institute of Marine Science, Townsville, Queensland, Australia
[4]AIMS@JCU, Australian Institute of Marine Science and James Cook University, Townsville, Queensland, Australia
[5]Department of Ecology and Evolutionary Biology, Pennsylvania State University, University Park, PA, USA
[6]Department of Ecology and Evolutionary Biology, University of California, Irvine, CA, USA
[7]Centre Scientifique de Monaco, 8 Quai Antoine 1er, Monaco, Monaco

JBL, 0000-0002-7005-9994; HAS, 0000-0002-2504-6836

Disease is an emerging threat to coral reef ecosystems worldwide, highlighting the need to understand how environmental conditions interact with coral immune function and associated microbial communities to affect holobiont health. Increased coral disease incidence on reefs adjacent to permanently moored platforms on Australia's Great Barrier Reef provided a unique case study to investigate environment–host–microbe interactions *in situ*. Here, we evaluate coral-associated bacterial community (16S rRNA amplicon sequencing), immune function (protein-based prophenoloxidase-activating system), and water quality parameters before, during and after a disease event. Over the course of the study, 31% of tagged colonies adjacent to platforms developed signs of white syndrome (WS), while all control colonies on a platform-free reef remained visually healthy. Corals adjacent to platforms experienced significant reductions in coral immune function. Additionally, the corals at platform sites that remained visually healthy throughout the study had reduced bacterial diversity compared to

healthy colonies at the platform-free site. Interestingly, prior to the observation of macroscopic disease, corals that would develop WS had reduced bacterial diversity and significantly greater community heterogeneity between colonies compared to healthy corals at the same location. These results suggest that activities associated with offshore marine infrastructure impacts coral immunocompetence and associated bacterial community, which affects the susceptibility of corals to disease.

## 1. Introduction

Coral disease has the potential to significantly reduce coral cover and diversity on reefs, and is predicted to be on the rise [1,2]. While coral disease prevalence is low on Indo-Pacific reefs, increasingly frequent localized disease events have been linked to local anthropogenic impacts including eutrophication, sedimentation, terrestrial pollution and increased ocean temperatures [3–7]. To understand connections between changing marine environments and disease, and to effectively manage disease outbreaks, studies that explore how environmental stressors affect both the immunocompetence of corals and the structure of coral-associated microbial communities are needed. However, holistic studies that simultaneously examine the interplay among environmental stressors, host physiology and associated microbes within coral reef ecosystems are lacking ([8], but see [9,10]).

Corals are complex organisms comprised of diverse and dynamic consortia of eukaryotes, prokaryotes and archaea, whose symbioses are essential to coral health [11–16]. Many coral-associated bacteria potentially complement important host functions, such as nitrogen fixation and sulfur cycling, that allow corals to thrive in oligotrophic waters [17–19]. Others protect their hosts from harmful pathogens by producing antimicrobial compounds and potentially disrupting the pathogens' cell-to-cell communication [20–22]. While coral-associated microbial communities can be beneficial to their hosts, changes in environmental and/or host conditions can dramatically shift microbial community structure, with potentially negative effects for the coral host. For example, in favourable habitats, corals host highly stable microbial communities, while those in unfavourable habitats host less structured and more diverse communities [23]. Furthermore, environmental perturbations can reduce the capacity of the host and/or the microbiome to regulate community composition, resulting in an unpredictable and unstable microbial community state [24]. For example, shifts in bacterial assemblages have been recorded on coral colonies exposed to thermal, nutrient, carbon and pH stress [25]. Ritchie [20] also noted a loss of antimicrobial activity within coral mucus during a warm thermal anomaly that coincided with a bleaching event, and Bourne *et al.* [26] reported microbial community shifts on healthy colonies prior to visual signs of bleaching.

The fate of the coral host appears to be linked to the diversity and composition of its associated microbial consortium. For example, bacterial community shifts have been proposed to be a primary driver of coral bleaching [27], and a number of other diseases including Aspergillosis [28], white plague and black band disease [29–31], have been linked to altered coral-associated microbial assemblages [32–35]. Paradoxically, it is still unknown if these changes arise as a cause or consequence of pathogenesis [36,37]. Nevertheless, shifts in coral-associated microbiota are emerging as a useful indicator of altered coral health state [38].

Corals also possess an innate immune system that helps protect them from infection by potentially harmful organisms. One of the best-studied components of the coral immune system is the prophenoloxidase-activating system, or melanization cascade, which involves the conversion of prophenoloxidase to phenoloxidase and the subsequent deposition of melanin following pathogen detection or injury [39]. Melanin helps seal wounds and forms a potent physico-chemical barrier that restricts the movement of invading organisms. In many invertebrate species, disease resistance and basal activity of the prophenoloxidase system are positively correlated with higher baseline activity levels, conferring greater disease resistance to individuals and populations [40–43]. In corals, activity of the prophenoloxidase system has been used as a proxy for immune function, and elevated activity of the melanization cascade has been linked to tissue damage during *in vitro* pathogen exposures and *in situ* coral infection [10,44–48]. While coral innate immunity clearly plays an important role in maintaining coral health on reefs, little is known about the potential influence of environmental stress on immunological pathways, how innate immune function changes during the entire course of disease progression (i.e. from initially healthy corals through disease infection and subsequent recovery or death), and how disease progression relates to microbial community structure.

Tourism on the Great Barrier Reef (GBR) represents a $6.4 billion per year commercial industry in Australia [49]. Tourism on the reef is geographically concentrated, with 85% of tourist activities focused in the Cairns and Whitsundays regions, and day trips to pontoons representing the largest component of the tourism industry [50,51]. Impacts of pontoon structures on the surrounding reef community may include damage from installation of moorings, shading of the reef, concentrated visitor activity (i.e. physical fin damage [52], short term water quality changes from sunscreen [53] and human waste [54]), and potential seabird guano runoff resulting in decreased water quality. While there is strong motivation for tourism operators to protect reefs associated with platforms, there is great complexity of coral response to stress, and previous work has identified that additional anthropogenic impacts associated with offshore marine tourist structures appear to overwhelm the immune system, resulting in increased immune gene expression and higher disease incidence [47].

Reports of localized increased disease prevalence at offshore reef platforms on the GBR [55] provided an opportunity to undertake a case study to holistically investigate the complex interplay that occurs among environmental stress, coral-associated microbial communities, coral immune function and disease onset *in situ*. We combine ecological disease monitoring, water quality assessment, protein-based immune function characterization and microbial community profiling using 16S rRNA gene amplicon pyrosequencing to build a baseline understanding of the interactions that occur between the coral host and its microbiota during environmental stress to culminate in disease. Specifically, we compare coral immune function and coral-associated microbial communities (i) between healthy corals adjacent to reef platforms versus healthy corals at a nearby control site, and (ii) among corals before, during and after initial signs of coral disease white syndromes (WSs).

# 2. Material and methods

## 2.1. Study site

The case study was conducted at Hardy Reef (19°44′33″ S, 149°10′57″ E), a mid-shelf reef situated 75 km offshore of the Whitsunday Island group in the central region of the Great Barrier Reef Marine Park (figure 1) from November 2010 to June 2011. Hardy Reef had one 45 m × 12 m platform which accommodated up to 400 visitors per day [50], and an unused smaller platform (24 m × 10 m) approximately 300 m south of the main tourist platform (figure 1). The smaller platform was previously used for tourism, however use of this platform ceased approximately 1 year prior to this study. Both platforms were permanently moored approximately 5 m from the reef crest, and provided tourists with snorkel and diving tours, air-conditioned lounges, and fresh water showers and toilet facilities. In addition to the two platform sites, we also monitored a control site situated 800 m to the south and down-current of the unused platform that had no permanent structures and received no tourists (figure 1).

## 2.2. Weather monitoring and water quality sample collection

The Whitsundays region of the GBR has a monsoonal climate, with a summer wet season from approximately December to March. Daily water temperature, rainfall and light intensity data were collected by the Australian Institute of Marine Science (AIMS) weather station located at the main tourist platform (data available from http://www.aims.gov.au). Means for these physical variables were calculated using daily values from a 14-day period including, and immediately preceding, each month's sampling date. Daily visitor data were not available and thus visitor concentration was unable to be included as a water quality covariate.

Beginning in January, five replicate water samples were collected in 50 ml sterile syringes approximately 1 m above the reef substrate at each sampling location and time point. Sub-samples were analysed for dissolved inorganic nutrients (ammonium, nitrite, nitrate, phosphate, silicate) and dissolved organic nitrogen (DON), phosphorus (DOP) and carbon (DOC). Five additional replicate samples were collected in 500 ml plastic bottles for salinity measurements using a Portasal Model 8410A Salinometer (Guildline, Ontario, Canada). Duplicate sub-samples were immediately filtered through 0.45 µm syringe filters (Sartorius MiniSart N, Goettingen, Germany) and collected in 10 ml acid-washed screw-cap tubes for dissolved nutrient quantification. DOC samples were acidified by adding 100 µl of analytical-grade hydrochloric acid (32%). All samples were immediately snap frozen in liquid nitrogen and stored at −30°C prior to laboratory analysis. Inorganic dissolved nutrient concentrations were determined by standard wet chemical methods [56] implemented on a segmented flow analyser [57]. Dissolved

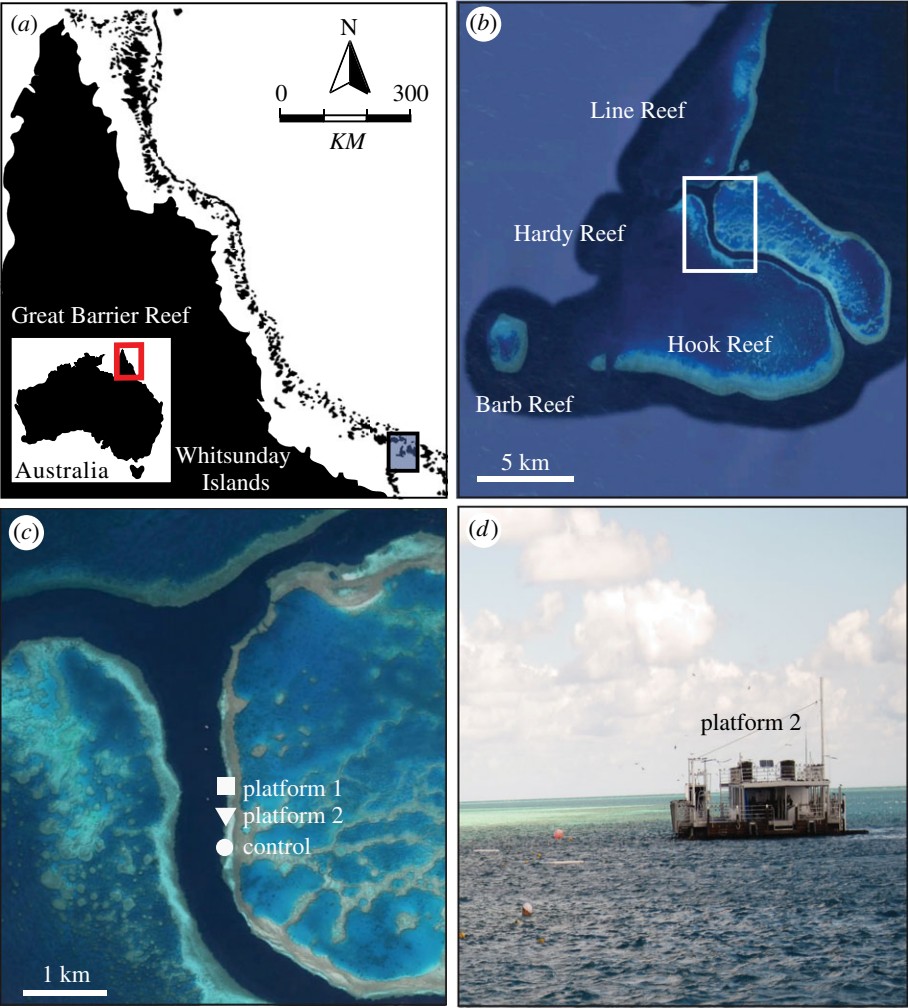

**Figure 1.** Map showing location of (*a*) study site at Hardy Reef, which lies 75 km offshore within the (*b*) central sector of Australia's Great Barrier Reef Marine Park. (*c*) Three coral monitoring and sampling sites: tourist platform, unused platform, and a control site. (*d*) The unused platform lies approximately 300 m south of the tourist platform and the control site lies an additional 800 m south of the unused platform 2. Aerial images: Google Earth.

inorganic nitrogen (DIN) was calculated by summing the separately measured ammonium, nitrite and nitrate values, and dissolved inorganic phosphorus (DIP) represents total phosphate. Analyses of total dissolved nitrogen and phosphorus (TDN and TDP) were carried out using persulphate digestion of filtered water samples [58], which were then re-analysed for inorganic nutrients, as described above. DON and DOP were calculated by subtracting DIN and DIP concentrations from the TDN and TDP values, respectively. DOC concentrations were measured by high temperature combustion (680°C), using a Shimadzu Total Organic Carbon TOC-5000A carbon analyser (Kyoto, Japan).

## 2.3. Coral monitoring and sample collection

At 2–3 m depth within each of the three locations monitored, eight similarly-sized (30–40 cm diameter) visually healthy colonies of the coral *Acropora millepora* were tagged with a plastic cattle tag, which was inscribed with a unique colony identification number and attached to the colony with a plastic cable tie. Care was taken to select colonies separated by a minimum of 5 m to minimize the risk of confoundment due to potential vector transmission of infectious disease agents. Tagged colonies were photographed and their health states visually assessed each month. Additionally, samples for bacterial community profiling and immunological analyses were collected at four time points: November (late austral spring), January (early austral summer, wet season), February (mid austral summer, wet season) and June (early austral winter, dry season). At each time point, one branch (approximately 5 cm in length) was sampled from the middle of each tagged colony using surgical bone cutters and placed in

a plastic bag on SCUBA. If a colony displayed signs of disease, an apparently healthy portion of a branch was collected approximately 1 cm from the disease lesion boundary, rather than the middle of the colony. All tagged colonies were photographed before and after sample collection. Coral samples were placed in 15 ml cryogenic tubes, snap-frozen in liquid nitrogen and stored at $-80°$C until processing. To attempt to minimize bias resulting from triggering an immune response as a result of physical injury from fragment removal [48] or bacterial proliferation, samples were collected in the same order during each monthly time period and frozen within 15 min of collection. Additionally, the lesion itself was not included in further sample processing as the initial PO response due to injury is local to the lesion [48].

## 2.4. Genomic DNA extraction and PCR amplification of bacterial 16S rRNA gene

Frozen coral fragments were crushed in a sterile, stainless steel, liquid nitrogen chilled mortar and pestle using a hydraulic press. Bacterial DNA was then extracted from 100 mg (wet weight) aliquots of the crushed coral powder using the PowerPlant DNA Isolation Kit (MoBio, Carlsbad, CA) according to the manufacturer's instructions. Purified DNA was stored at $-80°$C until PCR amplification. Library preparation and bacterial tag-encoded FLX-titanium amplicon pyrosequencing based on the V1–V3 region (*E. coli* position: 27–519) of the small-subunit ribosomal RNA (16S) gene was performed at MRDNA (Shallowater, TX) on all samples with forward primer 27F (GAGTTTGATCNTGGCTCAG) and reverse primer 519R (GTNTTACNGCGGCKGCTG), as described previously [59,60].

## 2.5. Sequence processing and selection of operational taxonomic units

Sequence reads were processed using the Quantitative Insights Into Microbial Ecology 1 (QIIME) pipeline, as described previously [61]. Briefly, samples were demultiplexed by sample-specific barcodes, low quality reads were discarded (minimum read length: 150 bp; maximum read length: 500 bp; minimum average Phred score: 25; maximum ambiguous bases: 6; maximum homopolymer run: 6; maximum primer mismatches: 0), and chimeric sequences were discarded (Chimera Slayer [62]). Operational taxonomic units (OTUs) were identified (method: Uclust, threshold: 97% [63]), and representative sequences were chosen by consensus and assigned a taxonomy (method: PyNAST [64], template: GreenGenes v13_8 [65]). A lane mask was applied to hide uninformative regions and a phylogenetic tree was constructed (method: FastTree [66]). Following processing, 148 182 classifiable, non-chimeric reads of sufficient quality remained, averaging 2554 reads per sample. Prior to downstream analysis, sequence data for all samples was rarefied to 685 reads to remove sequencing effort heterogeneity. These sequence data are available from the GenBank Sequence Read Archive (SRA) under accession number SRP148975.

## 2.6. Protein extract preparation and assays

Total potential phenoloxidase (tpPO; [48]) activity was measured as a proxy for innate immune function following the methods described by Palmer *et al.* [45], with minor modifications. Tissue was airbrushed from frozen coral fragments (approx. 4 cm$^2$) into 10 ml of ice-cold extraction buffer (50 mM Tris-HCl, pH 7.8 with 50 mM dithiothreitol) and homogenized for 45 s (IKA T10 Basic homogenizer, Malaysia). The resulting tissue slurry was centrifuged at 3500 rpm for 5 min and the supernatant collected and stored at $-30°$C until use. Total tissue protein content was determined using the DC Protein Assay (Bio-Rad, Hercules, CA, USA) according to the manufacturer's standard assay protocol. The assay was held at room temperature for 20 min and the endpoint absorbance at 750 nm was measured using a Spectramax M2 spectrophotometer (Molecular Devices, Sunnyvale, CA, USA).

Twenty microlitres of tissue extract from each sample was added in triplicate to a clear 96 well microtiter plate, followed by the addition of 40 µl of Tris buffer (50 mM, pH 7.8) and 25 µl of trypsin (0.1 mg ml$^{-1}$). For blanks, tissue extract was replaced by 20 µl of extraction buffer. After a 20-min incubation at room temperature, 30 µl of dopamine hydrochloride (10 mM, Sigma-Aldrich, St. Louis, MO, USA) was added. The kinetic absorbance at 490 nm was determined at 5-min intervals for 45 min. Total potential PO activity was calculated as the change in absorbance using the linear portion of the reaction curve over time, standardized to the total protein content of each sample.

## 2.7. Data analyses

To better tease apart the effects of location and health state, samples were separated into three groups on the basis of their proximity to artificial structures and visually-assessed health state during the study:

(i) colonies at the control site that remained visually healthy during the entire study period (control), (ii) colonies at the platform sites that remained visually healthy during the entire study period (platform) and (iii) colonies at the platform sites that developed signs of the coral disease white syndrome (WS) at any point during the study. No colonies of *A. millepora* at the control site displayed visual signs of WS at any point in the study.

Bacterial community (alpha) diversity was assessed using Faith's phylogenetic distance, a phylogenetic measure of diversity based on total branch length of the bacterial 16S rRNA gene phylogeny, calculated in QIIME 1 [67,68]. Beta diversity (between sample diversity) was assessed using the unweighted UniFrac distance metric (a phylogeny-based distance metric that avoids treating semi-quantitative pyrosequencing results as quantitative and is more sensitive to the influence of rare taxa in shaping coral-associated bacterial communities) [69]. A UniFrac dissimilarity matrix based on bacterial sequences at the OTU level was used to construct unconstrained two-dimensional principal coordinates analysis (PCO) plots to visualize differences between bacterial community assemblages [70,71]. Additionally, permutational multivariate analysis of variance (PERMANOVA; [70]) was used to test for statistical differences between: (i) bacterial community assemblages associated with colonies remaining visually healthy throughout the study period at platform versus control sites, and (ii) colonies located at the platform sites remaining visually healthy throughout the study versus those developing WS. This analysis was based on unweighted UniFrac dissimilarity matrices, type III partial sums of squares and 9999 random permutations of the residuals under the reduced model. *Post hoc* pair-wise comparisons among locations and health states were conducted when significant main effects were detected. Additionally, the similarity percentages routine (SIMPER) was used to investigate the contribution of individual bacterial OTUs to the observed separation between sample groups. The SIMPER analysis was based on a zero-adjusted Bray–Curtis similarity matrix [70] using presence/absence-transformed data at the lowest phylogenetic level (i.e. OTU level).

Differences in water quality variables (salinity, DIN, DIP, silicate, DON, DOP, DOC) among sampling locations and months, and interactions between these factors were determined by two-way, fixed-factor PERMANOVA using a Gower Metric-based distance matrix [71]. The factor 'Location' contained two levels (platform and control), and the factor 'Month' contained three levels (January, February and June). *Post hoc* pair-wise comparisons among locations and months were conducted when significant main effects were detected. The Gower Metric was deemed the most appropriate resemblance measure because the water quality variables employed in the analyses were on different scales, there were no zero values, and the various physical and chemical variables merited equal weighting (e.g. equal differences between values have the same influence on association, regardless of scale). To identify water quality variables responsible for driving differences among sampling locations for each month (those contributing the most to patterns in multivariate space), we used a PCO performed on the original Gower matrix [71]. Correlations of the ordination axes using the original water concentration data were overlaid as vectors on a bi-plot. All multivariate analyses were performed using Primer 6.0 statistical software (Primer-e Ltd, UK).

Mean total potential phenoloxidase enzymatic activity and bacterial community alpha diversity were analysed using univariate repeated measures analyses of variance across the four sampling months ('Time'), which was tested as the within-subject factor, and three 'Health State' groups (as described above: healthy near control, healthy near platform, and WS near platform), which was tested as the between-groups factor. Analyses were conducted on log-transformed data and the assumption of sphericity [72] was assessed prior to interpretation ($\alpha = 0.05$). All *post hoc* comparisons were performed using Tukey's honestly significant difference (HSD) analyses. Univariate analyses were performed using R v. 3.0.2 [73].

# 3. Results

## 3.1. Physical environment characteristics and colony condition

All tagged colonies monitored at the control site ($n = 8$) remained visually healthy throughout the duration of the study (figure 2$a$–$c$,$g$). All tagged colonies located adjacent to the two platforms ($n = 16$) were visually healthy at the beginning of the study in November and December (figure 2$g$). In January, five colonies located at platform sites (i.e. one at the large used platform and four at the small unused platform) displayed tissue loss exposing intact white skeleton, characteristic signs of WS. Lesions radiated from the centre of colonies as diffuse, acute to sub-acute areas of tissue loss,

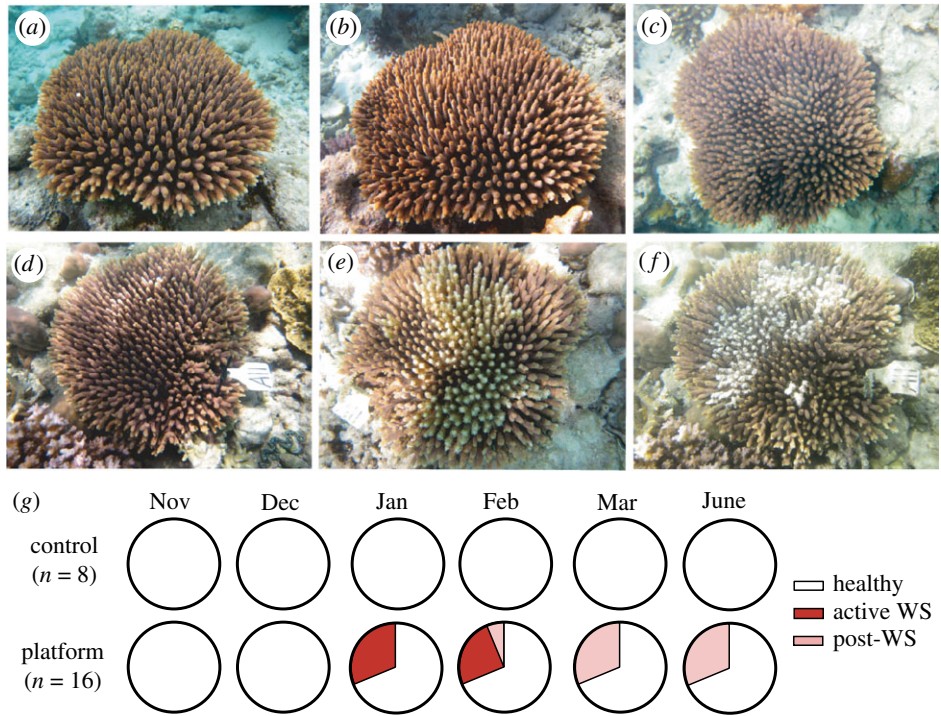

**Figure 2.** Time-series examples of two tagged colonies of *A. millepora*. Both colonies were visually healthy in November 2010 at the control site (*a*) and a platform site (*d*). Subsequent visual health status was assessed in January 2011 (*b*: apparently healthy at control site; *e*: WS signs at platform site) and the final visual health status was assessed in June 2011 (*c*: apparently healthy at control site; *f*: recovered from WS at platform site, but displaying partial-colony mortality). Time series of colony health status by month (*g*).

with no evidence of predation. WS-induced tissue loss represented mortality of 40% to 50% of the tissue surface area of affected colonies (figure 2*a,d*). Between December 1 and January 1, the reef experienced the greatest increase in mean daily water temperature ($+1.4°C$; electronic supplementary material, figure S1c) recorded throughout the eight-month study, as well as 7 consecutive days of rain that, when averaged, was greater than 1 standard deviation above the mean amount accumulated over the entire study period (mean $\pm$ s.d. $= 5.4 \pm 13.1$ mm d$^{-1}$ for the study, $20.4 \pm 9.6$ mm d$^{-1}$ over this 7-day period). In February, no new colonies developed disease signs and, apart from one colony at a platform site with progressing WS, all disease lesions had ceased progression and no characteristic WS bands were evident (figure 2*f,g*). Between January 26 and February 6, the reef experienced the highest mean daily wind speed, light intensity, water temperature and rain accumulation (electronic supplementary material, figure S1a–d). During this period, a severe tropical cyclone passed 270 km north of the Whitsunday region and the reef experienced 4 consecutive days of rain, representing 3 standard deviations above the study mean (3-day mean $\pm$ s.d. $= 63.2 \pm 16.6$ mm d$^{-1}$). No further disease development or lesion progression was observed in March, and all colonies previously recorded with WS at platform sites appeared visually healthy, with healed tissue margins around areas of partial colony mortality. By June, none of the 24 tagged colonies had succumbed to the disease, but rather all tagged colonies were again visually healthy.

## 3.2. Spatial and temporal patterns in water quality

Water quality parameters did not vary between the platform and control sites (pseudo-$F = 0.75$, $p = 0.61$), however they did vary significantly among sampling months (pseudo-$F = 19.0$, $p < 0.001$, figure 3, electronic supplementary material, tables S1 and S2). Seasonal differences in salinity and silicate strongly separated the wet austral summer months of January and February from the drier month of June along the first PCO axis (60.4% of total variation, figure 3 and electronic supplementary material, table S3). February was characterized by higher concentrations of dissolved inorganic nitrogen (DIN) and lower concentrations of dissolved organic carbon (DOC) and dissolved organic nitrogen (DON), resulting in the slight distinction between the two summer months along the second PCO axis (19.3% of total variation, figure 3; electronic supplementary material, table S1). There

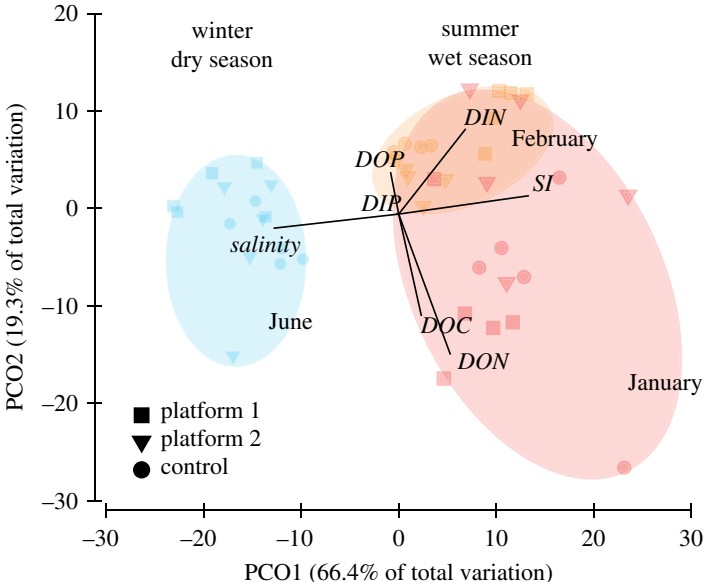

**Figure 3.** Two-dimensional principal coordinates plot visualizing relationships among concentrations of water quality variables and salinity during sampling months in the wet season denoted as clusters in January (red) and February (orange) and the dry season in June (blue) for each of the sampling locations. DIP, dissolved inorganic phosphorus; Si, silicates; DIN, dissolved inorganic nitrogen; DOP, dissolved organic phosphorus; DON, dissolved organic nitrogen; DOC, dissolved organic carbon. $n = 5$ replicate measurements per site per month.

was no significant interaction between month and sampling location (pseudo-$F = 3.5$, $p < 0.001$, figure 3, electronic supplementary material, table S1).

## 3.3. Anthropogenic influence on coral-associated bacterial communities and coral immune function

Amplicon sequencing resulted in 148 182 classifiable, non-chimeric reads of sufficient quality, with an average of 2554 reads per sample. Bacterial communities from all coral samples were dominated by Proteobacteria (70% of reads; mostly Xanthomonadales, 33% of reads) and Oceanospirillales (15% of reads; electronic supplementary material, figure S2). Bacterial communities associated with healthy colonies at each location (i.e. healthy colonies at platform sites versus healthy colonies at control site) fell into consistent phylogenetic clusters over the eight-month sampling period (d.f. = 1, pseudo-$F = 1.43$, $p = 0.03$, figure 4, electronic supplementary material, table S4). Clustering within location, visualized by a partitioning of the samples in the PCO ordination (figure 4a), indicates that healthy samples from platform sites were more similar to each other in bacterial phylogenetic structure than they were to healthy control site samples (d.f. = 1, pseudo-$F = 1.43$, $p = 0.03$, electronic supplementary material, table S4). While overall bacterial communities did not differ significantly between sampling time points (d.f. = 3, pseudo-$F = 1.04$, $p = 0.34$, electronic supplementary material, table S4), differences between locations were most pronounced in January (d.f. = 1, pseudo-$F = 1.58$, $p = 0.005$, figure 4b) and June (d.f. = 1, pseudo-$F = 1.32$, $P = 0.03$, figure 4c, electronic supplementary material, table S4).

While there was no significant interaction between location and time point, to further characterize microbial community dynamics resulting in disease, we examined fine-scale changes within each sampled time point. In January, a loss of bacterial taxa associated with corals from platform sites was identified as the major driver of bacterial community differences between healthy corals at platform versus control sites. Twenty-three out of the 24 (96%) OTUs that each explained greater than or equal to 1% of the separation between locations (identified by SIMPER) were less common on platform site corals (electronic supplementary material, table S3). The single OTU that was more common on platform site corals belonged to the genus *Burkholderia* and was present in 80% of platform corals compared to only 25% of control corals (electronic supplementary material, table S3). Furthermore, bacterial diversity on healthy platform site corals (mean phylogenetic diversity: $6.2 \pm 1.4$, mean ± s.e.) was nearly 50% lower than on corals at the control site ($11.8 \pm 0.3$) (HSD, $p = 0.008$, figure 5a). At the same time, bacterial communities associated with healthy platform corals became less stable between

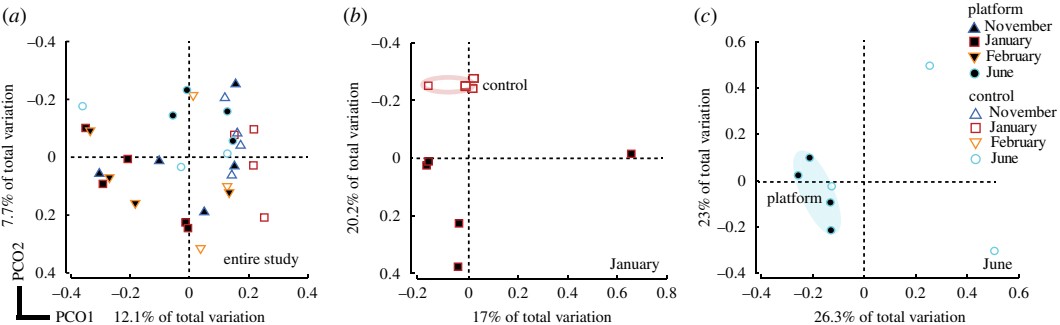

**Figure 4.** Two-dimensional principal coordinate ordination plots visualizing dissimilarity between bacterial communities (unweighted UniFrac distance) associated with colonies of the coral *Acropora millepora* remaining visually healthy throughout the study at control (white symbols) and platform sites (black symbols) (*a*) throughout the eight-month study, (*b*) in January only, and (*c*) in June only. Blue triangles: November samples; red squares: January samples; orange triangles: February samples; and blue circles: June samples.

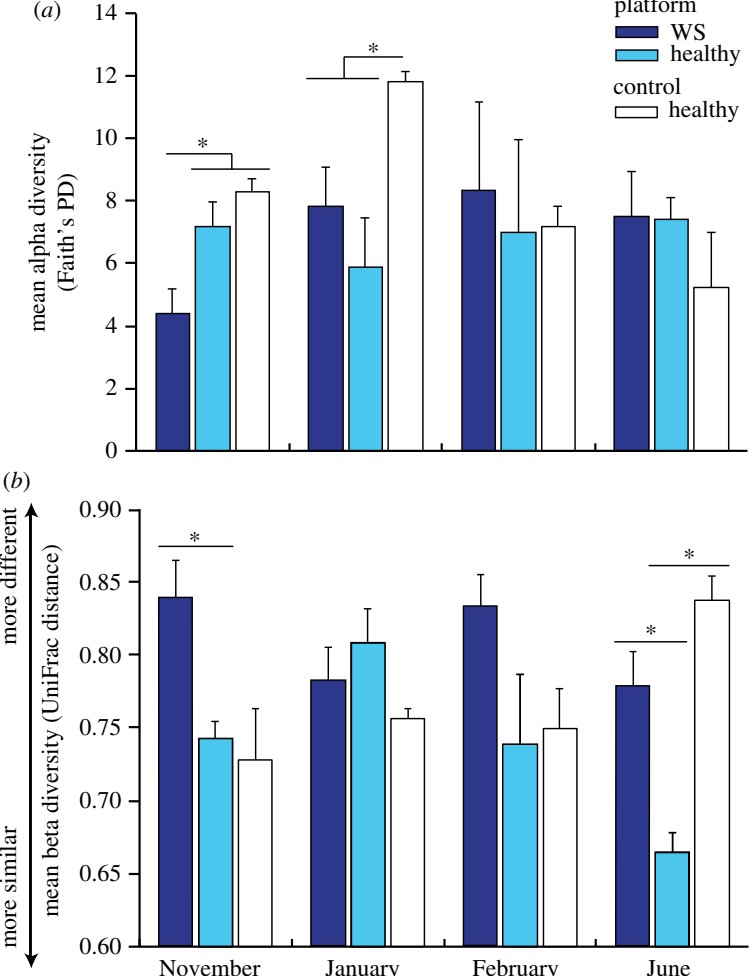

**Figure 5.** Comparisons of bacterial communities associated with *A. millepora* among healthy corals at control sites (white bars, *n* = 4 colonies), healthy corals at platform sites (light blue bars, *n* = 5 colonies), and colonies with WSs at platform sites (dark blue bars, *n* = 5 colonies) at four time points (November, January, February and June) for (*a*) mean alpha diversity (measured as Faith's phylogenetic distance); and (*b*) mean beta diversity (measured as UniFrac distance). A lower score in panel (*b*) indicates community composition is more similar between colonies, and conversely, a higher score indicates community composition is more dissimilar between colonies. Error bars: standard error of the mean. The asterisks denote a significant within-month difference in (*a*) phylogenetic diversity using Tukey's HSD; and (*b*) UniFrac distances using a *t*-test with 999 Monte Carlo permutations.

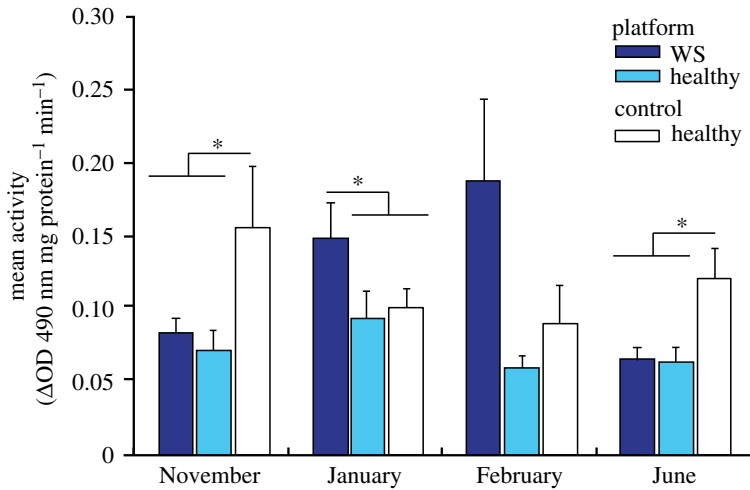

**Figure 6.** Mean total potential phenoloxidase activity ($\Delta$ absorbance mg protein$^{-1}$ min$^{-1}$) of apparently healthy colonies of *A. millepora* located near platforms (light blue bars) and the control site (white bars). Dark blue bars represent colonies located near the platform in November that would later present visual signs of white syndrome (WS) disease in January and February, and then subsequently recover (post-WS) in June. $n = 5$ replicate colonies per location and health state. Error bars represent the standard error of the mean.

individuals, which can be roughly visualized as the amount of within-location dispersion (variability) in the PCO ordination (figure 4*b*). However, community stability among healthy samples did not differ significantly between platform (mean within-location UniFrac distance: $0.81 \pm 0.02$) and control sites ($0.76 \pm 0.01$) (HSD, $p > 0.05$, figure 5*b*), suggesting that differences in bacterial community diversity (i.e. alpha diversity) rather than among-colony heterogeneity (i.e. beta diversity) underpins the dissimilarity detected between these locations.

In June, disparities in among-colony bacterial community composition (i.e. beta diversity) appeared to drive the separation observed between healthy corals at the two locations. However, in contrast to January samples, bacterial community structure was far more consistent (i.e. homogeneous) among healthy platform site corals (mean UniFrac distance $\pm$ s.e.: $0.66 \pm 0.01$) than control site corals (mean within-location UniFrac distance: $0.84 \pm 0.02$) (HSD, $p = 0.002$, figure 4*b*). These differences in bacterial community stability can be visualized as the large dispersion of control site samples relative to the more tightly clustered platform samples in the PCO ordination (figure 4*c*). Unlike patterns found in January, loss of specific bacterial taxa and reduced overall diversity did not appear to drive community differences. Nearly 30% (10 out of 37) of OTUs explaining greater than or equal to 1% of the separation between locations were more abundant at control sites (electronic supplementary material, table S5), and overall bacterial diversity did not differ significantly between locations (HSD, $p > 0.05$, figure 4*a*).

Total potential phenoloxidase activity levels ($\Delta$ absorbance mg protein$^{-1}$ min$^{-1}$) of healthy corals varied significantly between locations ($F = 5.7$, $p = 0.02$), but not among months ($F = 1.2$, $p = 0.33$), with a significant interaction between location and month ($F = 2.7$, $p = 0.028$). In both November and June, mean tpPO activity of healthy corals near reef platforms was approximately 50% lower than mean activity of corals at the control site (November: platform [$0.07 \pm 0.01$ s.e.] versus control [$0.15 \pm 0.04$ s.e.], HSD, $p = 0.021$; June: platform [$0.06 \pm 0.008$ s.e.] versus control [$0.12 \pm 0.02$ s.e.], HSD, $p = 0.034$; figure 6). However, mean tpPO activity did not differ between locations in January or February ($p > 0.05$; figure 6).

## 3.4. Influence of white syndrome on coral-associated bacterial communities and coral immune function

At platform sites, bacterial taxa present in communities associated with corals that developed WS in January did not differ significantly from those associated with corals remaining healthy over the course of the study (i.e. healthy colonies at platform sites versus WS-affected colonies at platform sites) (d.f. = 1, pseudo-$F = 1.16$, $p = 0.20$, electronic supplementary material, table S4) and no significant differences were detected among sampling time points (d.f. = 3, pseudo-$F = 1.13$, $p = 0.14$, electronic supplementary material, table S4). The lack of consistent clustering according to health state,

visualized as the absence of partitioning of the samples in the PCO ordination (electronic supplementary material, figure S3), indicates that overall, bacterial communities on corals developing WS were not different to communities on corals remaining healthy throughout the study.

Although consistent shifts in bacterial community membership were not detected between healthy and WS-affected corals, two key changes in bacterial community structure were observed in apparently healthy corals that would develop WS disease signs. In November, two months prior to the first recorded visual signs of disease, the diversity of bacteria associated with apparently healthy colonies that would subsequently develop WS (mean $\pm$ s.e.: $4.4 \pm 0.8$) was nearly 40% lower than on those remaining healthy ($7.2 \pm 0.8$, HSD, $p = 0.03$, figure 5a). Additionally, variation in bacterial communities among corals (i.e. beta diversity) was significantly greater on pre-WS colonies (mean UniFrac distance: $0.84 \pm 0.03$) relative to corals remaining healthy throughout the study (mean UniFrac distance: $0.74 \pm 0.01$; HSD, $p = 0.001$, figure 5b). These results indicate that bacterial communities on pre-WS corals were less diverse and more variable than on corals remaining healthy, even before the first visible appearance of macroscopic disease signs.

In June, following the cessation of disease progression, beta diversity (i.e. among colony community differences) was significantly higher for post-WS colonies (mean UniFrac distance $\pm$ s.e.: $0.78 \pm 0.02$) compared to samples of platform corals remaining healthy throughout the study (mean UniFrac distance $\pm$ s.e.: $0.66 \pm 0.01$; HSD, $p = 0.002$, figure 5b). However, bacterial diversity did not differ significantly between the two groups (HSD, $p > 0.05$, figure 5a).

In November, prior to the first visual signs of disease, tpPO activity did not differ significantly between colonies that would later develop WS (mean tpPO activity $\pm$ s.e.: $0.08 \pm 0.007$) and platform corals remaining healthy throughout the study ($0.07 \pm 0.01$; $p > 0.05$, figure 6). In January, when macroscopic disease signs were first observed, mean tpPO activity of WS-infected colonies ($0.15 \pm 0.02$) was slightly elevated relative to healthy colonies ($0.10 \pm 0.01$), but these differences were not significant (HSD, $p > 0.05$). In February, when all but one WS case had ceased progression, mean tpPO activity in WS-affected colonies ($0.18 \pm 0.06$) was threefold higher than in those remaining healthy ($0.06 \pm 0.007$; HSD, $p = 0.03$, figure 6). In June, when all WS cases had healed, mean tpPO activity of colonies that had previously suffered WS tissue loss did not differ significantly from mean activity of those that had remained healthy throughout the study ($0.07 \pm 0.009$ versus $0.06 \pm 0.008$, respectively; HSD, $p = 0.87$; figure 6).

# 4. Discussion

In this case study, we report significant reductions in immune function and loss of bacterial diversity on healthy corals adjacent to reef platforms relative to healthy corals at a platform-free control site. During the austral summer (January), levels of WS peaked at platform sites, affecting 31% of corals monitored, while all tagged corals at the control site remained visually healthy throughout the study. While there were no overall significant differences between bacterial community over the course of the study between diseased and healthy corals, fine-scale differences were detected over time. In November, two months before the first visual signs of disease, colony-level bacterial diversity associated with corals that would go on to develop disease signs in January was significantly lower relative to corals remaining healthy at platform sites. At the same time, bacterial communities became significantly more variable among corals that would become diseased compared to the more stable communities that characterized healthy corals at this time. This observation is consistent with previous coral microbiome studies that contributed to the postulated Anna Karenina principle, which states that stressors reduce the capacity of the host and/or their microbiomes to regulate microbial community composition leading to unstable and often stochastic community states [24]. These results suggest that proximity to reef platforms impacts coral immunocompetence and coral-associated bacterial community structure and diversity, which may in turn influence the susceptibility of corals to disease.

## 4.1. Reduced coral-associated bacterial diversity and community stability precede white syndrome coral disease

The significant reduction in bacterial diversity associated with individual corals and the significant increase in among-colony bacterial community heterogeneity in corals that developed WS, months before the first visual signs of disease, highlights the important role that diversity is likely to play in stabilizing microbial communities that govern coral health. WS levels at platform sites peaked during

the warm, rainy austral summer, affecting 31% of tagged platform site corals in January, while all tagged control site corals remained visually healthy throughout the study. The timing of this disease event corresponded with a period of marked microbial disturbance at platform sites, even among healthy corals. On both macroscopic and microscopic scales, biodiversity stabilizes ecological systems through functional redundancy and complementarity, with different species flourishing under different conditions and thereby buffering the impacts of environmental change [74–78]. On macroscopic scales, decreased species diversity often leads to elevated risk of abrupt and potentially irreversible ecosystem collapse [74,75,78–81]. At the microbial level, reduced bacterial diversity within mammalian organs (e.g. the intestine) is known to diminish the ability of individuals to resist infection, assimilate nutrients and maintain the aggregate function of a healthy microbiome [82–85]. While little is known about the mechanisms between coral-associated microbial diversity and coral health, an array of microbes is now recognized as essential to coral resilience [15,86]. The results presented here support that a loss of microbial diversity may impact coral holobiont resilience, an observation consistent with the Anna Karenina principle [24].

Coral-associated microbes contribute significantly to coral health through nutrient cycling, antibiotic production and disruption of pathogen-to-pathogen communication [17,18,20,21]. Diverse microbial assemblages are therefore likely to provide a high level of functional redundancy, helping to buffer the impacts of environmental perturbations and maintain coral health. However, when microbial diversity is suppressed, as we see in November prior to the first visual signs of WS, functional redundancy is potentially reduced, hindering the coral-associated microbial communities' ability to maintain physiological functions that are vital to the health of the coral host. Furthermore, increased between-coral microbial community heterogeneity immediately prior to developing macroscopic WS disease signs suggests a loss of stability in community structure that may underpin bacterial community resistance and/or resilience. Taken together, decreased microbial diversity on individual corals and elevated heterogeneity among corals prior to disease onset suggests that an overall disruption in microbial community structure, rather than infection by a single pathogenic species, could contribute to disease development [87]. We however cannot eliminate the possibility of an unidentified disease agent impacting these corals between November and January, initiating or potentially precipitating the transition from an apparently healthy to a diseased state.

## 4.2. Reduced coral-associated bacterial diversity and coral immune function adjacent to reef platforms

Reduced bacterial diversity and suppression of coral immune function in apparently healthy corals adjacent to reef platforms reveals a potential mechanism contributing to the observed 15-fold increase in coral disease levels near reef platforms previously reported by Lamb & Willis [55]. While the exact cause of these shifts could not be identified in the current study, it does not appear that nutrient inputs either directly or indirectly are associated with the reef platforms as patterns are more closely associated with seasonal differences. However, other potential factors influencing differences among platform and platform-free locations should be explored. On macroscopic ecosystem scales, persistent human disturbances are known to alter the stability and diversity of ecological systems [74,75,78–81] and microscopic bacterial communities are similarly sensitive to environmental perturbations [88]. While little is known about specific environmental drivers of coral-associated microbial diversity, evidence from soil microbiology studies indicates that both chemical and physical disturbance can significantly reduce bacterial diversity in disturbed soils [89,90]. Marine microbes on oligotrophic coral reefs are generally nutrient limited, so disturbances increasing the availability of nutrients are expected to play a similarly important role in shaping bacterial community structure [91,92]. Microbial community shifts have been recorded on colonies of *Porites cylindrica* exposed to nutrient-rich fish farm effluent [93], as well as corals experimentally exposed to elevated nutrient levels [25,94]. Microbial shifts associated with elevated iron concentrations adjacent to shipwreck sites have also been linked to coral mortality [95]. In addition, higher levels of microbial pollution, sediment and plastic waste have all been linked to coral disease [5,7,96], but microbial links between these factors and disease remain unidentified.

Disruptions in coral-associated community structure and the appearance of visual disease signs also coincided with significant changes in coral immune function. Previous observations of low baseline phenoloxidase activity and melanin levels among healthy Indo-Pacific acroporids suggest that when corals in this family are challenged with injury, pathogenic agents and/or environmental perturbation, they must significantly upregulate their immune function to avoid infection [6,10,43–48,97]. Since a

microbial shift was observed prior to visual disease signs, one might expect a corresponding immune response. However, the lack of preemptive immune system activation, and the previously reported reduced levels of components of the lectin-complement system (one of the most effective innate immune effector mechanisms) in these corals compared to those that remained visually healthy near the platforms [47] suggests corals affected were either unable to detect oncoming disease and/or were physiologically incapable of mounting a preemptive immune response. In January, corals responded to disease using TLR signalling and complement system activation along with an increase in phenoloxidase activities [47]. This response was likely sufficient to halt disease progression by February, in part due to the significant elevation of phenoloxidase activity among WS-affected corals when cessation of tissue loss associated with WS lesions was first observed. This suggests a potential role for phenoloxidase in WS cessation and/or wound healing, which is consistent with previous observations of elevated phenoloxidase levels at the border of WS lesions [45] and wound lesions [48]. Baseline levels of total potential phenoloxidase activity in control corals potentially conferred disease resistance, but could also reflect a response to a separate insult, and longer term data would be required to better establish baseline activity. However, not all platform site corals developed disease despite their relatively low tpPO activity levels, highlighting the intricate interplay between host, environmental and microbial factors that govern coral health and disease.

Recently, it was shown that corals near platforms had increased expression levels of genes involved in the Toll-like receptor (TLR) signalling pathway, particularly in summer [47]. As TLR signalling is crucial for the detection of microbes and the initiation of the immune response, and was shown to regulate the microbiome in cnidarians through anti-microbial peptide (AMP) secretion [98], the increased levels potentially indicate microbial stressors were present at these locations. However, the reduced levels of immune effectors, such as tpPO activity, could suggest that the antimicrobial defences were negatively impacted, contributing to disease development. In contrast, the baseline levels of tpPO activity in control site corals potentially conferred disease resistance. Although the exact mechanisms remain unknown, elevated nutrient levels have been shown to reduce immunocompetence in a range of marine invertebrate species [99–102]. For example, transcript levels encoding for prophenoloxidase decreased by 60% in blue shrimp (*Litopenaeus stylirostris*) exposed to elevated ammonia levels [99,103]. Similarly, reduced PO activity and immune cell counts were found in abalone (*Haliotis diversicolor supertexta*) under elevated ammonia concentrations. Abalone exposed to elevated nitrite levels also showed weakened immune function, characterized by reduced phagocytic activity and decreased pathogen clearing efficiencies, despite increased phenoloxidase activity [100]. These studies demonstrate that elevated nutrient levels can have deleterious effects on marine invertebrate immunocompetence, thereby increasing disease susceptibility, and suggest that similar phenomena could occur in corals.

The lack of significant differences in nutrient levels detected between platform and control sites in this study may reflect the relatively infrequent water quality sampling regime and the absence of November samples. This low-resolution dataset could have missed sporadic nutrient pulse events like seabird guano runoff associated with periodic rainfall. Reef platforms often accommodate large numbers of seabirds that deposit nitrogen, phosphorus and potassium-rich guano [104] that during periods of heavy rainfall, is washed from the platforms into the surrounding seawater. Additionally, pulses of high visitor concentrations could introduce short-term water quality changes from human waste or cause stress to corals through physical damage. Alternatively, other unidentified parameters could underlie the observed reductions in immune function and microbial diversity on corals adjacent to reef platforms. Future work would benefit from higher resolution water quality sampling combined with longer-term monitoring of more colonies over time. Identification of environmental drivers will be critical to the effective management of offshore reef platforms and should remain a research priority. Interestingly, disease incidence was higher at the unused platform than the platform where visitors are permitted, suggesting that the infrastructure itself, rather than human presence, may be responsible for inducing stress. Further study should evaluate if the patterns observed in this case study are consistent at reefs adjacent to other reef platforms or marine infrastructure, and should incorporate visitor concentration in analyses.

# 5. Conclusion

This study assessed the interplay among coral hosts, associated microbial communities and environmental drivers preceding and throughout a disease event that affected corals adjacent to offshore marine-based infrastructure. Here, we did not observe significant differences in microbial

community structure between diseased and healthy corals, however we did detect significant reductions in bacterial diversity and significantly more variable bacterial communities on corals that would develop WS at reef platforms, two months before the first visible signs of disease. The loss of microbial diversity on corals adjacent to reef platforms and also on corals prior to the development of disease signs suggests that microbial diversity plays an important role in the maintenance of coral holobiont function. We also found significant reductions in coral immune function and coral-associated bacterial diversity adjacent to reef platforms, even among corals remaining visually healthy throughout the eight-month study. While we were unable to identify the mechanistic driver(s) of these changes, such information can be critical for effective reef management. Importantly, corals have the potential for long lifespans, and thus the seven-month span of this study represents only a snapshot of disease dynamics. Hence population level disease surveys should be combined with the methods used herein (i.e. microbial and immunity analyses) to better detail the dynamics of disease at reefs with tourist platforms. In conclusion, these results indicate that activities associated with proximity to reef platforms (i.e. anthropogenic disturbance and/or nutrient enrichment from seabird guano runoff) impacts coral immunocompetence and coral-associated bacterial community structure and diversity, which affects a coral's susceptibility to disease.

Ethics. Samples were collected on permit numbers G07/23617.1 in 2010 and G11/34003.1 in 2011, issued by the Great Barrier Reef Marine Park Authority.

Data accessibility. The sequence data supporting this article are available from the GenBank Sequence Read Archive (SRA) under accession number SRP148975.

Authors' contributions. J.B.L., B.S. and B.L.W. conceived the study. J.B.L. and J.A.J.M.v.d.W. collected and processed water and coral samples with field assistants listed under Acknowledgements. J.B.L. and J.A.J.M.v.d.W. performed protein-based coral immunological assays. F.J.P. performed DNA extractions and sample preparation for genetic sequencing with laboratory assistance listed under Acknowledgements. D.G.B., J.B.L., F.J.P. and J.A.J.M.v.d.W. performed bioinformatics and statistical analyses. All authors contributed to writing the manuscript.

Competing interests. The authors declare no competing interests.

Funding. Funding for this study was provided by an Australian Institute of Marine Science and James Cook University (AIMS@JCU) research award awarded to J.B.L. and in-kind support from Fantasea Adventure Cruising. Funding was also provided by the Australian Research Council Discovery Project DP0451196 awarded to B.L.W.

Acknowledgements. We thank L. Kelly, P. Cross, S. Beveridge, S. Harte, C. Heeres, V. Barry Dale, E. Smart, and T. Heintz for their assistance in sample collection and logistical support. We thank Sefano M Katz for laboratory assistance.

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
