## [Reviewer comments · Royal Society Open Science]

Review History

RSOS-190355.R0 (Original submission)

Review form: Reviewer 1

Is the manuscript scientifically sound in its present form?

Yes

Are the interpretations and conclusions justified by the results?

Yes

Is the language acceptable?

Yes

Is it clear how to access all supporting data?

Yes

Do you have any ethical concerns with this paper?

No

Have you any concerns about statistical analyses in this paper?

No

Recommendation?

Accept with minor revision (please list in comments)

Comments to the Author(s)

I enjoyed reading this manuscript. It is well organized and well written. My major concern with this study is the short temporal scale and the potential for confounding factors that might have affected colonies before the study began or during the study. A temporal "replication" (longer study of the same colonies) would help clarify many patterns, especially for organisms that are long lived. This is a limitation in many studies of these modular, long lived organisms. A short paragraph in the discussion stating the limitations of the study (sampling, spatial and temporal scales, platform replication, etc. etc. would be informative and a nice complement to the work.

The higher disease prevalence near the unused platforms is similar to results and observations in the Caribbean where, higher number of diseases and disease prevalence is often found in areas away (better conditions) from human impacted areas.

A little more information on what sort of human activities are carried out in the platforms and surroundings might be helpful.

Review form: Reviewer 2 (Laurie Raymundo)

Is the manuscript scientifically sound in its present form?

Yes

Are the interpretations and conclusions justified by the results?

Yes

Is the language acceptable?

Yes

Is it clear how to access all supporting data?

Yes

Do you have any ethical concerns with this paper?

No

Have you any concerns about statistical analyses in this paper?

No

Recommendation?

Accept with minor revision (please list in comments)

Comments to the Author(s)

Please see attached file (Appendix A).

Decision letter (RSOS-190355.R0)

24-Apr-2019

Dear Ms Smith

On behalf of the Editors, I am pleased to inform you that your Manuscript RSOS-190355 entitled "Reduced diversity and stability of coral-associated bacterial communities and suppressed immune function precedes disease onset in corals" has been accepted for publication in Royal Society Open Science subject to minor revision in accordance with the referee suggestions. Please find the referees' comments at the end of this email.

The reviewers and handling editors have recommended publication, but also suggest some minor revisions to your manuscript. Therefore, I invite you to respond to the comments and revise your manuscript.

- Ethics statement

- Data accessibility

<http://datadryad.org/submit?journalID=RSOS&manu=RSOS-190355>

- Competing interests

- Authors' contributions

- Acknowledgements

- Funding statement

Because the schedule for publication is very tight, it is a condition of publication that you submit the revised version of your manuscript before 03-May-2019. Please note that the revision deadline will expire at 00.00am on this date. If you do not think you will be able to meet this date please let me know immediately.

- 1) A text file of the manuscript (tex, txt, rtf, docx or doc), references, tables (including captions) and figure captions. Do not upload a PDF as your "Main Document";
- 2) A separate electronic file of each figure (EPS or print-quality PDF preferred (either format should be produced directly from original creation package), or original software format);
- 3) Included a 100 word media summary of your paper when requested at submission. Please ensure you have entered correct contact details (email, institution and telephone) in your user account;
- 4) Included the raw data to support the claims made in your paper. You can either include your data as electronic supplementary material or upload to a repository and include the relevant doi

within your manuscript. Make sure it is clear in your data accessibility statement how the data can be accessed;

5) All supplementary materials accompanying an accepted article will be treated as in their final form. Note that the Royal Society will neither edit nor typeset supplementary material and it will be hosted as provided. Please ensure that the supplementary material includes the paper details where possible (authors, article title, journal name).

on behalf of Dr Denise Greig (Associate Editor) and Kevin Padian (Subject Editor)
openscience@royalsociety.org

Associate Editor Comments to Author (Dr Denise Greig):

This study brings together a lot of data in a natural setting with a Before After Control Impact study design. I appreciate all the work that went into interpreting these data sources and the interplay between environmental factors, microbial communities, disease, and immunity. The changes in bacterial diversity prior to disease development are fascinating. It is well written, but can be tough to follow in places because there is so much disparate information to absorb. Both reviewers had concerns about the coral monitoring sample sizes and I am wondering if you can offer some justification/rationale for the number of quadrats you used at each site as well as the distances between them (i.e. how independent were they? and how representative of that area as a whole?).

Additionally, I think the terminology gets confusing in some places obscuring the results. One example is on line 79 when you use the word "stochastic". I think this is part of the "Anna

Karenina principle" that you mention in more detail in the discussion, but in this sentence it is unclear if the perturbations are stochastic or the microbial community response is stochastic. If the microbial community response is random, then to me, this implies that disease, immunity and environmental factors would not matter. Maybe the sentence at lines 453-456 should be moved up to the introduction. At the very least, I would move the sentence at lines 453-456 to earlier in the discussion when you first mention the "Anna Karenina principle" and clarify the meaning of the sentence in the intro.

One of the reviewers had issues with the meanings for diversity versus heterogeneity. Alpha and beta diversity, and the tests used for each, were nicely described in the methods, but it got confusing in the results section. Specifically, I was confused about the interplay between diversity and heterogeneity in lines 348-357. I don't know if additional subheadings would help here, maybe by season? Or if you just need to go through and make sure you are using the same terms consistently.

Line 418-432. In this paragraph, I am confused about the immune finding described in the first sentence and the way it is summed up at the end of this paragraph (and also in the abstract). The sentence cites significant reductions in immune function on platform v control corals, but looking at Figure 5, the phenoloxidase activity is identical between apparently healthy coral at the platform and control sites in Jan and Feb when the WS coral appears to mount a healthy immune response (although I guess the response was quite variable in Feb)... what if the Jan and Feb phenoloxidase activity levels are baseline for the control site and in November and June, the control site corals were responding to some separate insult?

I like supplementary Figure 1 and because it is the first thing you refer to in the Results, I would consider including it in the main portion of the manuscript.

For Figure 4, I don't quite understand the y-axis label. The caption says it is within-category distance...which categories are being referred to here?

Reviewer comments to Author:

Reviewer: 1

Comments to the Author(s)

I enjoyed reading this manuscript. It is well organized and well written. My major concern with this study is the short temporal scale and the potential for confounding factors that might have affected colonies before the study began or during the study. A temporal "replication" (longer study of the same colonies) would help clarify many patterns, especially for organisms that are long lived. This is a limitation in many studies of these modular, long lived organisms. A short paragraph in the discussion stating the limitations of the study (sampling, spatial and temporal scales, platform replication, etc. etc. would be informative and a nice complement to the work.

The higher disease prevalence near the unused platforms is similar to results and observations in the Caribbean where, higher number of diseases and disease prevalence is often found in areas away (better conditions) from human impacted areas.

A little more information on what sort of human activities are carried out in the platforms and surroundings might be helpful.

Reviewer: 2

Comments to the Author(s)

Please see attached file.

Author's Response to Decision Letter for (RSOS-190355.R0)

See Appendix B.

Decision letter (RSOS-190355.R1)

08-May-2019

Dear Ms Smith,

I am pleased to inform you that your manuscript entitled "Reduced diversity and stability of coral-associated bacterial communities and suppressed immune function precedes disease onset in corals" is now accepted for publication in Royal Society Open Science.

on behalf of Dr Denise Greig (Associate Editor) and Kevin Padian (Subject Editor)
openscience@royalsociety.org

Appendix A

Review of manuscript RSOS-190355

By L. Raymundo

This manuscript reports a study examining changes in microbial surface communities and immune function in diseased and healthy corals growing in areas exposed to tourism activity vs. those in an area remote from tourism across two seasons. The authors noted a decrease in immune capacity associated with tourism sites, as well as increased community heterogeneity and reduced diversity among surface bacterial communities.

The work was carefully carried out and adds to the body of knowledge regarding the physiological basis for coral responses to disease and environmental stress. As such, it should be published. Specific comments that the authors hopefully find useful are listed below:

Line 63: there should be a comma between “disease” and “and” and outbreak should be plural.

Line 115: considering what has already been published on this subject, the authors may wish to clarify that “concentrated visitor activity” involves both physical damage to corals and at least short-term impacts to water quality from both sunscreen and human waste. Also, I’m not quite sure how potential runoff can result from offshore pontoons.

Lines 136 and 147: the sample period is stated twice and is a bit redundant. Best to choose to mention in one section, but not both. It may also be informative to add some information on how visitor number may have varied within the sample period. Rain and temperature were discussed, but not visitor concentration.

Line 142: Please clarify whether the control site was upstream or downstream of the pontoons, in terms of current patterns.

Line 150: commas are needed around “and immediately preceding”

Line 173: I suggest moving “at 2-3 m depth” to the beginning of the first sentence: “At 2-3 m depth within each of the three locations monitored....”

Lines 180-182: I am not sure where this might fit best, but how did you control for-or at least consider-the potential response of corals to the physical injury resulting from branch removal for sampling? Because immune responses are known to be triggered as a result of physical injury, you might want to mention somewhere how you controlled for that (i.e., you treated all colonies the same way, presumably), and perhaps include a short reference to this in your discussion.

Line 263: Change “perMANOVA” to “PERMANOVA”?

Results: My main issue here is the very small sample size (8 tagged colonies per site) and the small number of diseased colonies that were subsequently sampled. A single colony at the used platform developed WS, as compared to none in the control site. Four colonies developed WS in the unused platform, which is 50% of that tagged population, but still a very small sample size. If the colonies were close together, and WS was caused by an infectious agent and transmitted by a vector (both are strong possibilities), then that could explain the results independent of the presence or absence of the

platform. While I appreciate the enormous amount of effort that went into this study, I think making statements suggesting an effect of the platforms could be made stronger if there were monitoring or assessment data of the larger population that could be cited here. I know that Lamb and Willis demonstrated significant differences in prevalence between platforms and control sites. Are there data available that were collected during this study of this larger population that could support these findings?

Likewise, the discussion (Line 438 and on) would also benefit from a reference to this; “31% of all platform site corals” does not adequately represent the population of platform corals, when this statement only considers 8 out of ? colonies of *A. millepora*.

Results Section on Anthropogenic Influences: This section (and subsequent mentions), I think, would benefit from a clarification of the differences between diversity and heterogeneity, as well as how “sample” is defined (i.e., one branch per colony? Or all branches from a single colony, with comparisons thus being made between colonies?). Much of the critical fine-scaled results discussed make reference to these differences. I’m assuming that sample heterogeneity means greater differences in abundance, as well as the total number of taxa/OTUs counted, but it would help to have that more clearly spelled out.

Last paragraph of the Discussion: Have the authors considered the pulses of tourists on the used platform as being a short-term water quality and physical damage impact? Seabirds are mentioned, so why not tourists?

Appendix B

Re: Response to Reviewers for Manuscript ID RSOS-190355

Reviewer: 1

I enjoyed reading this manuscript. It is well organized and well written. My major concern with this study is the short temporal scale and the potential for confounding factors that might have affected colonies before the study began or during the study. A temporal "replication" (longer study of the same colonies) would help clarify many patterns, especially for organisms that are long lived. This is a limitation in many studies of these modular, long lived organisms. A short paragraph in the discussion stating the limitations of the study (sampling, spatial and temporal scales, platform replication, etc. etc. would be informative and a nice complement to the work.

The authors are grateful for the constructive feedback from Reviewer 1. We have added a number of points within the discussion as suggested by the reviewer. We have not added a whole paragraph of limitations but incorporated these limitations in specific relevant points of the discussion and replicated below with corresponding line numbers in the final manuscript:

Line 520: "Future work would benefit from higher resolution water quality sampling combined with longer-term monitoring of more colonies over time."

Line 584: "Importantly, corals have the potential for long life-spans, and thus the 7-month span of this study represents only a snapshot of disease dynamics. Hence population level disease surveys should be combined with the methods used herein (i.e. microbial and immunity analyses) to better detail the dynamics of disease at reefs with tourist platforms."

The higher disease prevalence near the unused platforms is similar to results and observations in the Caribbean where, higher number of diseases and disease prevalence is often found in areas away (better conditions) from human impacted areas.

A little more information on what sort of human activities are carried out in the platforms and surroundings might be helpful.

More information has been added to detail the activities carried out at the platforms, as below from line 143:

"...and provided tourists with snorkel and diving tours, air-conditioned lounges, and fresh water showers and toilet facilities."

Reviewer 2: Prof Laurie Raymundo

This manuscript reports a study examining changes in microbial surface communities and immune function in diseased and healthy corals growing in areas exposed to tourism activity vs. those in an area remote from tourism across two seasons. The authors noted a decrease in immune capacity associated with tourism sites, as well as increased community heterogeneity and reduced diversity among surface bacterial communities.

The work was carefully carried out and adds to the body of knowledge regarding the physiological basis for coral responses to disease and environmental stress. As such, it should be published. Specific comments that the authors hopefully find useful are listed below:

The authors thank Prof. Raymundo for her thoughtful and thorough review of our work. We have attempted to respond to each comment below.

Line 63: there should be a comma between “disease” and “and” and outbreak should be plural.

Changed as suggested.

Line 115: considering what has already been published on this subject, the authors may wish to clarify that “concentrated visitor activity” involves both physical damage to corals and at least short-term impacts to water quality from both sunscreen and human waste. Also, I’m not quite sure how potential runoff can result from offshore pontoons.

Added as suggested, and clarified that runoff refers here to seabird droppings; line 116:

“...concentrated visitor activity (i.e. physical fin damage [52], short term water quality changes from sunscreen [53] and human waste [54]), and potential seabird guano runoff resulting in decreased water quality.”

Lines 136 and 147: the sample period is stated twice and is a bit redundant. Best to choose to mention in one section, but not both. It may also be informative to add some information on how visitor number may have varied within the sample period. Rain and temperature were discussed, but not visitor concentration.

Thank you for pointing out this redundancy. We have deleted the sentence from the second section (line 147).

Unfortunately we do not have access to visitor numbers to the Hardy Reef platform for the study period. Though monthly overall tourist visitation data to the entire Whitsundays region is available through GBRMPA, this data represents a region wide number not

specific numbers to the study site. While these data may reflect general trends in visitation, we are wary of incorporating these values as a covariate in the water quality analysis as they would not be robust in any way.

We have included the following text in the methods to acknowledge this limitation, line 153:

“Daily visitor data were not available and thus visitor concentration was unable to be included as a water quality covariate.”

And further text in the discussion, line 569:

“... and should incorporate visitor concentration in analyses.”

Line 142: Please clarify whether the control site was upstream or downstream of the pontoons, in terms of current patterns.

Clarified as suggested, see line 145:

“...we also monitored a control site situated 800 m to the south and down-current of the unused platform that had no permanent structures and received no tourists.”

Line 150: commas are needed around “and immediately preceding”

Completed as suggested.

Line 173: I suggest moving “at 2-3 m depth” to the beginning of the first sentence: “At 2-3 m depth within each of the three locations monitored....”

Revised as suggested.

Lines 180-182: I am not sure where this might fit best, but how did you control for-or at least consider the potential response of corals to the physical injury resulting from branch removal for sampling?

Because immune responses are known to be triggered as a result of physical injury, you might want to mention somewhere how you controlled for that (i.e., you treated all colonies the same way, presumably), and perhaps include a short reference to this in your discussion.

We have added the below text to indicate we did consider this potential bias, and attempted to control for it through rapid and consistent processing, line 191:

“To attempt to minimize bias resulting from triggering an immune response as a result of physical injury from fragment removal [48] or bacterial proliferation, samples were collected in the same order during each monthly time period and frozen within 15 minutes of collection. Additionally, the lesion itself was not included in further sample processing as the initial PO response due to injury is local to the lesion [48].”

We also added a reference, van de Water et al. 2015, The coral immune response facilitates protection against microbes during tissue regeneration [48] which discusses in detail the immune response as a result of physical injury.

Line 263: Change “perMANOVA” to “PERMANOVA”?

Fixed as suggested.

Results: My main issue here is the very small sample size (8 tagged colonies per site) and the small number of diseased colonies that were subsequently sampled. A single colony at the used platform developed WS, as compared to none in the control site. Four colonies developed WS in the unused platform, which is 50% of that tagged population, but still a very small sample size. If the colonies were close together, and WS was caused by an infectious agent and transmitted by a vector (both are strong possibilities), then that could explain the results independent of the presence or absence of the platform. While I appreciate the enormous amount of effort that went into this study, I think making statements suggesting an effect of the platforms could be made stronger if there were monitoring or assessment data of the larger population that could be cited here. I know that Lamb and Willis demonstrated significant differences in prevalence between platforms and control sites. Are there data available that were collected during this study of this larger population that could support these findings?

We acknowledge the limitations of our sample size, and have attempted to frame the work as a case study to be used as a launching point for further work. We have added text to detail that tagged colonies were collected from 5m apart to minimise the risk of vector transmission confounding our results, lines 179:

“Care was taken to select colonies separated by a minimum of 5m to minimize the risk of confoundment due to potential vector transmission of infectious disease agents.”

We have also made the limitation of our sample size more explicit, line 586:

“Hence population level disease surveys should be combined with the methods used herein (i.e. microbial and immunity analyses) to better detail the dynamics of disease at reefs with tourist platforms.”

Unfortunately, while Lamb and Willis did collect disease prevalence data from Hardy Reef, they did not monitor the same control site and therefore this data can not be included. However, our study aimed to determine the mechanism behind this finding, rather than to further demonstrate the difference in disease prevalence in the population, see line 486:

“Reduced bacterial diversity and suppression of coral immune function in apparently healthy corals adjacent to reef platforms reveals a potential mechanism contributing to the observed 15-fold increase in coral disease levels near reef platforms previously reported by Lamb & Willis [55].”

Likewise, the discussion (Line 438 and on) would also benefit from a reference to this; “31% of all platform site corals” does not adequately represent the population of platform corals, when this statement only considers 8 out of ? colonies of *A. millepora*.

Changed to *“31% of tagged platform site corals”*

Results Section on Anthropogenic Influences: This section (and subsequent mentions), I think, would benefit from a clarification of the differences between diversity and heterogeneity, as well as how “sample” is defined (i.e., one branch per colony? Or all branches from a single colony, with comparisons thus being made between colonies?). Much of the critical fine-scaled results discussed make reference to these differences. I’m assuming that sample heterogeneity means greater differences in abundance, as well as the total number of taxa/OTUs counted, but it would help to have that more clearly spelled out.

Diversity refers to within-coral bacterial community diversity (i.e. alpha diversity), whereas among-coral differences in community composition is referred to as heterogeneity (i.e. beta diversity). We have edited the text throughout to attempt to clarify this distinction, and have used the wording “more/less stable” in place of “heterogeneous,” and have been more explicit that these comparisons are among individuals.

Last paragraph of the Discussion: Have the authors considered the pulses of tourists on the used platform as being a short-term water quality and physical damage impact? Seabirds are mentioned, so why not tourists?

This is an excellent point, and we have added this into the text, see line 558:

“Additionally, pulses of high visitor concentrations could introduce short-term water quality changes from human waste, or could cause stress to corals through physical damage.”

Associate Editor Comments to Author, Dr Denise Greig:

This study brings together a lot of data in a natural setting with a Before After Control Impact study design. I appreciate all the work that went into interpreting these data sources and the interplay between environmental factors, microbial communities, disease, and immunity. The changes in bacterial diversity prior to disease development are fascinating. It is well written, but can be tough to follow in places because there is so much disparate information to absorb.

Both reviewers had concerns about the coral monitoring sample sizes and I am wondering if you can offer some justification/rationale for the number of quadrats you used at each site as well as the distances between them (i.e. how independent were they? and how representative of that area as a whole?).

We have added text to indicate that the corals selected for tagging were separated by a minimum distance of 5 meters in order to minimise potential bias due to either sampling of the same genet or by being within the dispersal distance of potential infectious disease vectors.

Unfortunately, one determinant for sample size was project budget, as the sequencing costs of greater than 24 samples across 4 time points would have been prohibitive. Fortunately, sequencing costs continue to become more affordable, and thus future work could easily increase the sampling effort. Additionally, we were limited by the tourism operator and GBRMPA as to the number of colonies that were allowed to be tagged. Eight colonies were permitted at the main tourist pontoon, and thus in the interest of consistency we tagged eight colonies at each of the three sites. We have not detailed the budget limitation nor the operator's imposed limitation in the manuscript, although we are amenable to adding a sentence if the Editor suggests doing so.

Additionally, I think the terminology gets confusing in some places obscuring the results. One example is on line 79 when you use the word "stochastic". I think this is part of the "Anna Karenina principle" that you mention in more detail in the discussion, but in this sentence it is unclear if the perturbations are stochastic or the microbial community response is stochastic. If the microbial community response is random, then to me, this implies that disease, immunity and environmental factors would not matter. Maybe the sentence at lines 453-456 should be moved up to the introduction. At the very least, I would move the sentence at lines 453-456 to earlier in the discussion when you first mention the "Anna Karenina principle" and clarify the meaning of the sentence in the intro.

We have attempted to clarify the sentence in the introduction, line 79:

"Furthermore, environmental perturbations can reduce the capacity of the host and/or the microbiome to regulate community composition, resulting in an unpredictable and unstable microbial community state."

We have also moved the sentence in the discussion as suggested, see lines 439.

One of the reviewers had issues with the meanings for diversity versus heterogeneity. Alpha and beta diversity, and the tests used for each, were nicely described in the methods, but it got confusing in the results section. Specifically, I was confused about the interplay between diversity and heterogeneity in lines 348-357. I don't know if additional subheadings would help here, maybe by season? Or if you just need to go through and make sure you are using the same terms consistently.

We have edited the text throughout to be more explicit about alpha versus beta diversity, please see tracked changes from lines 344-410.

Line 418-432. In this paragraph, I am confused about the immune finding described in the first sentence and the way it is summed up at the end of this paragraph (and also in the abstract). The sentence cites significant reductions in immune function on platform v control corals, but looking at Figure 5, the phenoloxidase activity is identical between apparently healthy coral at the platform and control sites in Jan and Feb when the WS coral appears to mount a healthy immune response (although I guess the response was quite variable in Feb)... what if the Jan and Feb phenoloxidase activity levels are baseline for the control site and in November and June, the control site corals were responding to some separate insult?

We have added in acknowledgement of this possibility, line 526:

“Baseline levels of total potential phenoloxidase activity in control corals potentially conferred disease resistance, but could also reflect a response to a separate insult, and longer term data would be required to better establish baseline activity.”

Small clarifications have also been made:

Line 517: *“...in these corals compared to those that remained visually healthy near the platforms...”*

Line 520: *“...corals responded to disease using TLR signaling and complement system activation along with an increase in phenoloxidase activities.”*

Line 529: *“However, not all platform site corals developed disease despite their relatively low tpPO activity levels...”*

Line 537: *“However, the reduced levels of immune effectors, such as tpPO activity, could suggest that the antimicrobial defenses were negatively impacted, contributing to disease*

development. In contrast, the baseline levels of tpPO activity in control site corals potentially conferred disease resistance.”

We have also moved this entire block of text to a later section to better fit within the manuscript framework.

I like supplementary Figure 1 and because it is the first thing you refer to in the Results, I would consider including it in the main portion of the manuscript.

This figure has been moved to the main manuscript as suggested.

For Figure 4, I don't quite understand the y-axis label. The caption says it is within-category distance...which categories are being referred to here?

The y-axis shows that lower scores indicate that microbial community composition is similar between samples of the same type (i.e. WS / healthy platform / healthy control). The figure has been clarified to show that part A is alpha diversity (mean Faith's PD) and part B is beta diversity (mean UniFrac distance). The caption has been updated to read:

“Figure 5: Comparisons of bacterial communities associated with A. millepora among healthy corals at control sites (white bars, n = 4 colonies), healthy corals at platform sites (light blue bars, n = 5 colonies), and colonies with WSs at platform sites (dark blue bars, n = 5 colonies) at four time points (November, January, February, and June) for (A) mean alpha diversity (measured as Faith's phylogenetic distance); and (B) mean beta diversity (measured as UniFrac distance). A lower score in panel (B) indicates community composition is more similar between colonies, and conversely, a higher score indicates community composition is more dissimilar between colonies. Error bars: standard error of the mean. The asterisks denote a significant within-month difference in (A) phylogenetic diversity using Tukey's HSD; and (B) UniFrac distances using a t-test with 999 Monte Carlo permutations”